# Developing and Validating the Narcissistic Personality Scale (NPS) among Older Thai Adults Using Rasch Analysis

**DOI:** 10.3390/healthcare9121717

**Published:** 2021-12-13

**Authors:** Pitchapat Chinnarasri, Nahathai Wongpakaran, Tinakon Wongpakaran

**Affiliations:** Department of Psychiatry, Faculty of Medicine, Chiang Mai University, Chiang Mai 50200, Thailand; pitchapat_chi@cmu.ac.th (P.C.); tinakon.w@cmu.ac.th (T.W.)

**Keywords:** narcissism, item response theory, reliability, validity, elderly, measurement, tool

## Abstract

Background: Being older could be stressful, especially among people with narcissistic personality disorders. Nevertheless, the tool is yet to be available among older Thai individuals. The study aimed to develop a tool to detect symptoms of narcissistic personality, and to validate its psychometric properties among older Thai adults. Methods: The Narcissistic Personality Scale (NPS) was developed based on nine domain symptoms of narcissistic personality disorder from the Diagnostic and Statistical Manual of Mental Disorders, Fifth Edition (DSM-5), consisting of 80 items. The original scale was field-tested using Rasch analysis for item reduction, rendering a final 43 items. NPS was further investigated among 296 seniors aged 60 years old. Rasch analysis was used to assess its construct validity. Result: Of 43 items, 17 were further removed as infit or outfit mean square >1.5. The final 26-item NPS met all necessary criteria of unidimensionality and local independence without differential item functioning due to age and sex, and good targeting with subjects. Person and item reliability were 0.88 and 0.95, respectively. No disordered threshold or category was found. Conclusions: The NPS is a promising tool with a proven construct validity based on the Rasch measurement model among Thai seniors. This new questionnaire can be used as outcome measures in clinical practice.

## 1. Introduction

People suffering from narcissistic personality disorder (NPD), according to DSM-5 diagnostic criteria for PDs, have a grandiose sense of self-importance (in fantasy or behavior), need for admiration, and lack of empathy [1]. This personality trait increases as individuals become older [2]. NPD is one of the personality traits that is difficult to treat. Prevalence of lifetime NPD was up to 6.2%, with rates greater for men (7.7%) than women (4.8%) [3]. Patients with NPD constantly deal with problems resulting from behaviors related to their personality. People with narcissistic personality usually value beauty, strength, and youth, thus aging creates unpleasurable conditions that are difficult for them to accept. It becomes evident that they may be more vulnerable to midlife crises than other groups [2]. The etiology of narcissistic personality disorder is multifactorial. Genetic predisposing is suggested to be one of the causes in some studies [4]. Aggression, reduced tolerance to distress, and dysfunctional affect regulation is prominent among individuals with NPD [5]. Developmental experiences, negative in nature, being rejected as a child, and a fragile ego during early childhood may have been linked to the occurrence of NPD in adulthood [6,7]. Evidence has suggested that the two distinct dimensions of narcissism, are often referred to as grandiose and vulnerable narcissism [8,9,10].

Individuals with NPD later in life may feel threatened when encountering declines in their health, beauty, and physiological, cognitive, and intellectual abilities, whereas normal older adults may adjust well to these changes. In addition, they might experience shame or vulnerability resulting from this threat to their autonomy [2]. As a result, older adults with NPD may experience some difficulty asking for assistance, leading to an increased difficulty handling their daily life activities. Based on feelings of entitlement to special privileges or status, they may act in inappropriate ways when requesting necessary assistance. Their narcissistic attitudes and behaviors may limit the number of family members available or willing to help. Evidence shows that relatives may have distanced themselves over the years due to interpersonal conflicts and tensions that frequently accompany NPD [2]. Seniors with NPD may suffer from problems regarding various life domains such as relationships, work, community connections, support, and interpersonal difficulties [11]. Such difficulties, accompanied by their fragile self-esteem, give rise to a susceptibility to varying mental disabilities such as substance use, mood, and anxiety disorders.

The prevalence of mood, anxiety, and substance use disorders totaled 17.4, 15.2, and 11.8%, respectively, among respondents with NPD [3]. Other personality disorders such as antisocial personality disorder, borderline personality disorder, histrionic personality disorder, and schizotypal personality disorder are also commonly comorbid among people with NPD [12]. A study showed that both vulnerability and grandiosity types of narcissism were significantly correlated with various personality disorders, except for schizotypal PD, as well as the personality traits of negative affect and antagonism [13]. Studies on narcissistic personalities among Thais are scarce. One conducted among young Thai athletes revealed that narcissistic admiration was positively correlated with self and coach-ratings of mental toughness, whereas narcissistic rivalry was negatively correlated with self and coach-ratings of mental toughness [14].

NPD can be diagnosed by psychiatrists and well-trained psychiatric or mental health professionals using structured interviews such as the Structured Clinical Interview for DSM IV Axis II Personality Disorder (SCID-II) [15,16]. Screening for NPD symptoms can be carried out using self-report questionnaires such as the Personality Diagnosis Questionnaire-4th Edition Plus (PDQ-4+) [17], Narcissistic Personality Questionnaire(NPQ), [18] and the Narcissistic Personality Inventory 40 (NPI-40) [19]. The NPI-16 and NPI-13 are the shorter versions of the NPI. They constitute widely used measurements of narcissism [20,21]; however, they are not specific to old age populations.

A review of the literature on personality disorders (PD) in late life revealed fewer research papers than those found for PD among younger adults. This could be due to the problem in providing diagnoses for late life personality disorders as well as age-related issues, e.g., changes in cognitive and social functioning and the effects of comorbid illness. All of these may complicate the diagnostic process [22].

However, lack of assessment of NPD among older adults may lead to less detection of NPD, resulting in less opportunity to prevent mental health problems from occurring. To the best of our knowledge, a specific tool to assess narcissistic personality among older adults is unavailable. More importantly, narcissistic personality may be expressed differently according to culture [23]. Moreover, despite the fact that a measurement should adhere to the standard DSM criteria, it would be vital that the measurement items are customized based on the respondent’s culture. The objectives of this study were to develop a culturally adapted tool called Narcissistic Personality Scale (NPS) based on DSM-5 criteria to detect the severity of symptoms of narcissistic personality, and to examine the NPS’s construct validity among Thai older adults.

## 2. Materials and Methods

This study used a cross-sectional online survey of older adults throughout Thailand between April and May 2021. To participate in the online survey, participants must read the protocol and accept an informed consent document on the first page of the questionnaire. Participants who refused to give their informed consent document were directed to the end of the survey. According to ethics principles, no respondents were forced to participate and could withdraw at any time. The study was approved by the Ethics Committee of the Faculty of Medicine at Chiang Mai University, Thailand.

### 2.1. Participants

The participants were older Thai adults residing in Thailand, aged 60 years and older, able to read and write in Thai, and able to access the Internet. The exclusion criteria included being diagnosed with schizophrenia, bipolar disorder, drug, or alcohol use disorder, and being intoxicated with alcohol within 24 h before participating in the study. After providing informed consent, participants were asked to complete the questionnaires on the Internet using personal computers, laptops, smartphones, or tablets.

### 2.2. Sample Size

As recommended by Linacre, a sample size between 30 and 200 could be sufficient for Rasch analysis [24]. A sample size of 30 could provide a statistically stable measure for the original draft of the questionnaire to reduce the items by identifying those misfitting. A sample size of at least 200 was used to further test for the second set of the questionnaires based on Rasch’s required criteria.

### 2.3. Instruments

#### Development of the Narcissistic Personality Scale (NPS)

The NPS was developed by constructing items based on nine domain symptoms of NPD DSM-5 diagnostic criteria [1]. The authors reviewed literature regarding the signs and symptoms of NPD based on DSM-5 criteria as well as from other diagnostic criteria other than DSM to generate the questions [25]. We also collected other clinical features from interviews with patients with NPD, and the patients’ key informant. We performed a focus group discussion regarding item selection, then consulted experts having experience with NPD to check for face validity. Finally, we obtained the draft of NPS consisting of 80 items, 23 items from domain 1 (grandiose sense of entitlement), 14 items from domain 2 (preoccupied with unlimited success, power, brilliance, beauty ideal love), 7 items from domain 3 (believes that he or she is “special” and unique), 8 items from domain 4 (need for admiration), 5 items from domain 5 (sense of entitlement), 7 items from domain 6 (interpersonally exploitative), 7 items from domain 7 (lack of empathy), 6 items from domain 8 (envy of others), and 4 items from domain 9 (shows arrogance). The NPS draft was then examined for content validity from the experts, i.e., three psychiatrists and one psychologist. The content validity index (CVI) was calculated to identify content validity quantitatively among the experts. A CVI ≥ 0.8 was considered acceptable [26]. The results were 0.75–1 for item CVI and 0.89 for scale CVI. The first draft was investigated using Rasch analysis in the field test, including 34 participants aged 60–89 years (mean age 70.76 ± 6.23 years), 55.9% were male and 44.1% were female. We excluded four participants because of protocol violation. The initial Cronbach’s alpha for 80 items of NPS was 0.96. Rasch analysis results showed 37 misfitting items that had mean square > 1.50; therefore, only 43 items were retained in the questionnaire. A few ambiguous items were corrected for this second draft version.

The 43-item NPS employs a 4-likert scale (highly agree = 3, moderately agree = 2, slightly agree = 1, disagree = 0). Total possible scores range between 0 and 78, with higher scores indicating greater narcissistic personality symptoms. Item samples include, “Many people approach you because you are a model of a successful person”, “Sometimes it becomes necessary to affect others to complete your job”, and “Occasionally people presume to act equal to me”.

The 43-item NPS was further analyzed in a larger sample size to acquire the fitting items for the final scale and to also determine its reliability and the validity of the narcissistic construct.

### 2.4. Statistical Analysis

For sociodemographic and descriptive statistics, mean, SD, and frequency were used. Internal consistency of the instrument using IBM SPSS, Version 22 (IBM Corp., Armonk, NY, USA) was determined using Cronbach’s alpha for which a value > 0.8 was considered acceptable.

Rasch analysis

The Polytomous Rasch rating scale model using the Winsteps Measurement Software (Winsteps, Rasch Measurement, Version 5.1.5.0, Chicago, IL, USA) verified the analyses.

#### 2.4.1. Examining the Fit between the Data and the Model

Chi-square fit statistics were calculated to indicate how well the empirical data fit the Rasch model. These fit statistics are the outlier-sensitive fit statistics mean square (outfit MNSQ) and information-weighted fit statistics mean square (infit MNSQ). The outfit statistic is more sensitive to outliers, whereas the infit MNSQ is sensitive to unexpected responses near the person’s ability level. The expected infit or outfit mean square values are 1.0; MNSQ > 2.0 indicates distorting or degrading the measurement system; MNSQ, 1.5–2.0 indicates unproductive for constructing measurement but not degrading. An item with infit or outfit MnSq beyond the 0.7–1.5 range was considered a misfit [27].

#### 2.4.2. Dimensionality Examination

Principal component analysis of the residual was used to identify the Rasch dimension, the only dimension in the data. However, a secondary dimension suggested the unexplained variance of the 1st contrast > 2 eigen values (at least 3). However, if the disattenuated correlation between person measures was more than 0.70, it may merely have been due to noise or an idiosyncratic item [28]. Local independence is a basic assumption of item response theory models in which the observed items are presumably independent of each other given an individual score on the latent variable(s). This is evaluated by determining a positive value of the correlation of size of the standardized residuals for two items (or persons). An acceptable correlation is less than 0.2, denoting that the pair of items is not duplicated or shared in the construct [29].

#### 2.4.3. Reliability and Separation Indices

Reliability is estimated both for persons and for items. Person reliability in Rasch analysis is comparable to Cronbach’s alpha. The higher the separation index, the better the instrument is able to differentiate person ability and item difficulty. Low person reliability indicates a narrow range of person measures or a small number of items. Low item reliability denotes that the sample is insufficient to locate the items on the latent variable correctly.

#### 2.4.4. Wright Map

Well targeting between person and item is denoted by the mean location for the person and should be around zero logits. It has been suggested that the difference between the mean value of the mean person measure should be within one logit. Floor or ceiling effects could also be examined visually using the map.

#### 2.4.5. Differential Item Functioning

We tested the differential item functioning (DIF) across sex, age, and education. Both statistical test and DIF contrast were used, and a DIF contrast > 0.64 indicated a substantial DIF [28].

#### 2.4.6. Category Function

Category functioning is assessed by determining category frequencies, mean measures, thresholds, and category fit statistics [30]. The items of the NPS have four categories. At least 10 responses per category is recommended for stable rating scale–structure threshold parameter estimates [30]. The mean measures and the thresholds should increase from lower to higher categories. An increase between 1.4 and 5 logits denotes a suitable threshold.

## 3. Results

### 3.1. Descriptive Analysis

The total number of responses was 326. Among them, 30 were excluded: 26 respondents were younger than the inclusion criteria (60 years old), and 4 were repeat responses. The final number of participants totaled 296, with 166 females (56.1%) and a mean age of 68.60 (SD = 7.53).

All baseline characteristics for older adults included in this study are shown below in Table 1.

### 3.2. Fit between the Data and the Model

Table 2 shows the item characteristics of all 43 items. All scores ranged between 1 and 4, with mean range between 1.47 and 3.48. Skewness ranged between −0.986 and 1.777, whereas kurtosis ranged from −1.063 to 2.357. All fell within acceptable ranges (<±2) [31].

All 43 items were analyzed using Rasch analysis. The results indicated 17 misfitting items in which the infit or outfit mean square was larger than 1.5, thus they were removed from the scale. All fit statistics for the remaining 26 items are shown in Table 3. The logit (measure) ranged between −1.00 and 0.84.

The fit values fell between 0.76 and 1.15 for infit MNSQ and 0.72 and 1.37 for outfit MNSQ. In exploring the dimensionality, the 26-item NPS was unidimensional indicated by the unexplained variance in the first contrast of 2.15 eigen values (less than 3), whereas the deattenuated correlation between the persona measure was 0.75 (>0.7). All of the largest standardized residual correlations were less than 0.2 indicating local independence.

### 3.3. Reliability

Person separation was 2.73 for the 26-item NPS and person reliability was 0.88, corresponding to Cronbach’s alpha of 0.93. The item–total correlation of 26 items ranged from 0.460–0.734. The McDonald’s ω coefficient was 0.933. This suggested three measurably distinct strata of participants were demonstrated on inner strength with the NPS. Item separation was 4.52 for the 26-item NPS and item reliability was 0.95, indicating that the item difficulty order was reproducible for this set of items for these subjects and that this sample was sufficiently large for this analysis. The NPS items appeared to be targeted well with the persons.

### 3.4. Wright Map

Figure 1 shows the person–item map for the NPS, which was deemed a good fit as the item mean was less than 1 logit. However, the item mean was slightly higher than the person mean, indicating that the items were more difficult. Some people with fewer traits of narcissistic personality might not be able to be assessed.

### 3.5. Differential Item Functioning (DIF)

The DIF contrast values ranged between 0.00 and 0.45 due to sex, and between 0.02 and 0.57 due to age. The DIF contrast values ranged between 0.00 and 0.76 due to education, item NPD08 (Often some people are jealous of you.) showed a DIF contrast of 0.66, whereas item NPD25 (If your child is not as good as you expected then you consider the child is not yours.) showed a DIF contrast of 0.76 (Table 3).

We followed up examining the DIF using the two stage DIF recommended by Zenisky, et. al. [32] by giving items 8 and 25 a weight of zero and reanalyzed. This will show the DIF for all items (including 8 and 25) but using the purified scoring. The DIF contrast yielded 0.71 for item 8 and 0.81 for item 25, suggesting DIF did not occur by chance.

### 3.6. Category Function

Figure 2 shows the category probability curves for an item of the 26-item NPS. No evidence of disordered thresholds with the four-category response was observed. This four-category response appeared to be appropriate.

## 4. Discussion

This study aimed to develop a validated questionnaire based on DSM-5 and culture for older adults using Rasch measurement theory as well as investigating possible item bias due to sex, age, and years of education. Rasch analysis results provided evidence that the 26-item NPS is a qualified tool based on Rasch measurement theory, i.e., it demonstrated unidimensionality, local independence, and acceptable fit statistics, indicating that all 26 items measured the same construct of narcissistic personality. The NPS revealed good reliability and acceptable targeting on the person–item Wright map, even though those with such low levels of narcissistic traits may not be sufficiently covered by the NPS. This could be because the items were generated mainly from the DSM criteria which is determined to capture the severe form of narcissism. Easier items might be added if we need the measurement to identify people with milder levels of a trait.

The 26-item NPS tends to have some item bias according to education, in that those who obtained higher levels of education tended to score higher on these two items despite the latent level of narcissistic trait being the same. These two items with DIF could potentially be removed from the scale. However, as suggested by Linacre, a large sample of 1000 may be required to confirm that real bias. To the best of our knowledge, no study has reported such information, even though the bias can be found in other personality measurements such as age bias in antisocial personality [33] or ethics bias in the schizotypal personality [34].

Narcissistic personality factor structure is an important issue to be noted. The 26-item NPS is drawn from eight of nine dimensions. The only dimension that is not included is “shows arrogance, haughty behaviors or attitudes”, which may be difficult to be captured using a self-reporting format. This contrasted with related research that points out the dimensional problems. Some investigators have questioned its one dimensionality and suggested separating the subscale, for example, grandiosity and entitlement [35]. However, our findings support the unidimensional construct of the NPD factor using the DSM-IV and DSM-5 symptoms documented by Miller et al. [36].

Other measurements, especially narcissistic personality inventory (NPI), which has been refined and revised to many versions, have been shown to have multiple components, and it has been suggested that the NPI subscale scores should not be totaled for an overall measure of narcissism [37]. Compared with the 26-item NPS, they have been shown to have a sufficiently unidimensional construct despite the fact that all items are from eight different dimensions. Therefore, the sum score of the NPS can be used to represent the degree of narcissism.

Culture plays important roles in the character of narcissism that would influence which items are to be selected for the questionnaire. A study comparing narcissism between western (Germany) and eastern (Japan) cultures documented that grandiose narcissism was prevalent in the western, whereas vulnerable narcissism was prevalent in the eastern country. The entitlement factor was assumed to be equivalent to the construct, whereas the relationship between narcissism and mental health problems differ [23]. We assumed that this cultural incongruency hypothesis concerning narcissism and mental health might be related to the different components of specific features of narcissism between the two cultures. A study conducted in Thai culture revealed how people express themselves through social media such as Selfies on Facebook, implying a narcissistic tendency. Older people with narcissistic personality might show their photos with family members expressing gratitude or care to inflate their self-esteem [38].

## 5. Strength and Limitation of the Study

The 26-item NPS constitutes, to the best of our knowledge, the first questionnaire developed considering older-aged people and collectivistic culture. However, the latter issue could also be seen as a potential limitation to generalizing the questionnaire to western or individualistic cultures. Replication studies in those cultures are required. Some limitations of the study involved the omission of some groups of people who experienced mental health problems such as bipolar disorder and drug or alcohol use disorder. In addition, some older people in Thailand may not be able to access the Internet, hence different results could probably be obtained if they were tested with the traditional paper and pencil method due to the difference of education, social status, income, place of residence, and the use of social media between those who were able to access and those who were not. Assessing people with low (i.e., normal) narcistic attitudes reliably is difficult because assessment is based on the DSM criteria, thus easier items that represent these criteria should be added and tested in further study.

Even though the small sample size was indicated as sufficient by the item reliability value, it prevented us from being confidently reassured regarding differential item functioning. A larger sample size should be warranted for future research. New items involving the “arrogant” dimension are still needed to cover the whole nine dimensions of DSM-5.

## 6. Implication and Future Research

The 26-item NPS can be used for outcome measures in evaluating the level of personality and can be used as an independent variable in association with other interested mental health outcomes such as depression, loneliness, anxiety disorder, or substance abuse. However, other psychometric properties, for example, convergent and discriminant validity, concurrent validity with other related measurements, and test–retest reliability, should be further examined. As the 26-item NPS is based on DSM, most items measure grandiose narcissism, and few item address vulnerable or covert narcissism, for example, item 21 (You do not think that you are good.). In addition, to make the scale better targeted to older populations, easier items prioritized to be added may concern vulnerable (covert) narcissism such as “be absorbed in thinking about his/her own affairs” or “be annoyed when other people ask for time and sympathy”. Moreover, the screening ability for narcissistic personality disorder against standard diagnostic tools can be further conducted to yield the cut-off score for the NPS to determine sensitivity, specificity, positive predictive value, and negative predictive value.

## 7. Conclusions

The NPS was developed through a rigorous process using both qualitative and quantitative methods. Input from participants, key informants, and experts were used for item generation. The questionnaires with content validity were tested and the number of items was reduced using Rasch analysis. The final 26-item NPS met the criteria based on Rasch measurement theory. It can be used in research and clinical practice to assess narcissistic personality.

## Figures and Tables

**Figure 1 healthcare-09-01717-f001:**
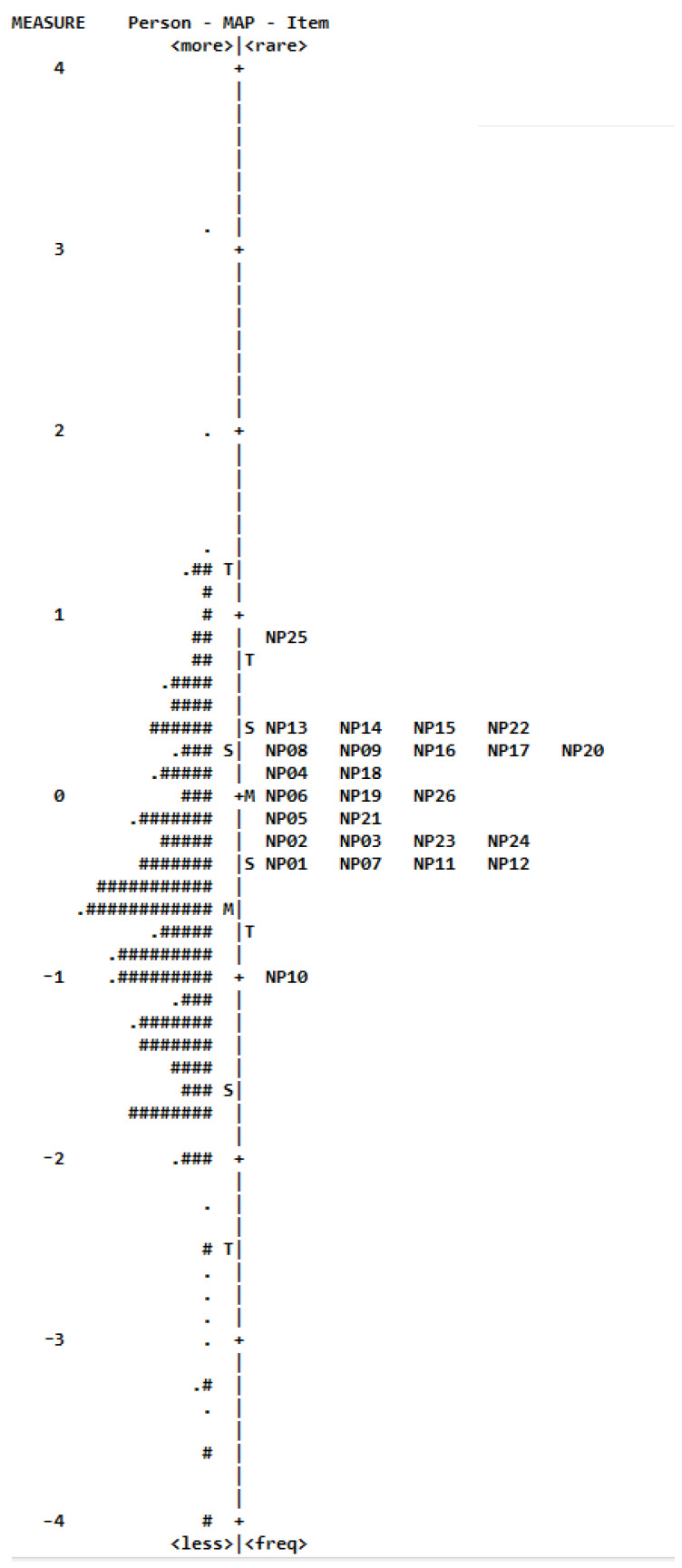
Person-item Wright Map. The persons are on the left of the vertical line, and items are located on the right of the vertical line. More able (narcissism) persons are located at the top of the map. More difficult (severe) items are located at the top of the map. Each “#” represents 2 persons. Each “.” represents 1 person (M = mean; S = 1 standard deviation from the mean; T = 2 standard deviations from the mean).

**Figure 2 healthcare-09-01717-f002:**
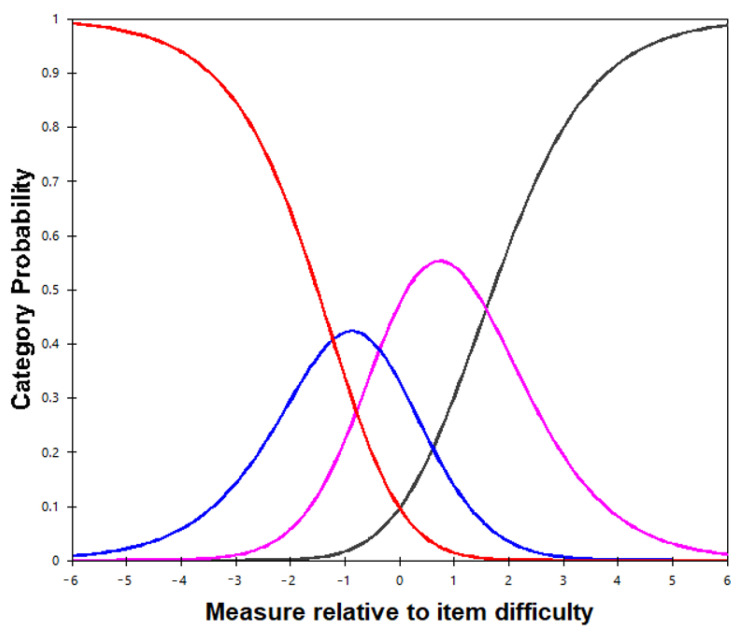
Category probability of the NPS. Notes: The curves for the NPS illustrate the range over which each of the four categories is most likely to be chosen. The red, blue, pink, and black curves on the graph represent the 4, 3, 2 and 1 NPS rating categories.

**Table 1 healthcare-09-01717-t001:** Sociodemographic data.

Characteristic	n (%) or Mean (S.D.)
Age (years), Mean (S.D.)	68.60 (7.53)
Sex: Female, n (%)	166 (56.1%)
Educational level, n (%)	
Higher than bachelor’s degree	46 (15.5%)
Bachelor’s degree	55 (18.6%)
Vocational/Diploma	22 (7.4%)
High school	48 (16.3%)
Elementary	112 (37.8%)
No education	12 (4.1%)
Missing	1 (0.3%)
Marital status, n (%)	
Single	25 (8.4%)
Married	200 (67.6%)
Widowed	61 (20.6%)
Divorced	10 (3.4%)
Income per month (THB)	
0–5000	102 (34.5%)
5001–15,000	93 (31.4%)
15,001–25,000	29 (9.8%)
25,001 and higher	72 (24.3%)
Occupation, n (%)	
General employee/Freelance	69 (23.3%)
Government officer	86 (29%)
Merchant/Business owner	38 (12.8%)
Unemployed/retired	67 (22.6%)
Other/unspecified	36 (12.1%)

S.D. = standard deviation, THB = Thai Baht.

**Table 2 healthcare-09-01717-t002:** Distribution of NPS items.

Item	Median	Mean	S.D.	Skewness	Kurtosis
NPS01	4	3.48	0.600	−0.776	0.144
NPS02	3	3.40	0.592	−0.507	−0.071
NPS03	3	2.99	0.804	−0.501	−0.160
NPS04	3	3.12	0.856	−0.695	−0.250
NPS05	1	1.87	1.057	0.838	−0.672
NPS06	3	2.43	0.928	−0.061	−0.884
NPS07	3	2.80	0.881	−0.354	−0.550
NPS08	2	1.91	0.998	0.755	−0.598
NPS09	2	2.10	0.981	0.379	−0.984
NPS10	3	3.25	0.867	−0.986	0.179
NPS11	2	2.28	0.973	0.150	−1.021
NPS12	3	2.74	1.025	−0.281	−1.060
NPS13	2	1.94	0.991	0.681	−0.696
NPS14	2	2.25	0.971	0.127	−1.063
NPS15	3	2.45	0.937	−0.084	−0.899
NPS16	3	2.90	1.042	−0.567	−0.866
NPS17	2	1.85	0.990	0.860	−0.449
NPS18	2	2.27	1.010	0.271	−1.025
NPS19	1	1.47	0.820	1.777	1.357
NPS20	2	2.25	1.023	0.317	−1.030
NPS21	2	2.33	0.986	0.145	−1.017
NPS22	2	1.96	0.991	0.627	−0.790
NPS23	2	2.05	0.900	0.493	−0.556
NPS24	2	2.32	1.008	0.190	−1.058
NPS25	2	2.01	0.928	0.533	−0.669
NPS26	2	1.80	0.885	0.959	0.176
NPS27	2	1.93	0.969	0.685	−0.632
NPS28	2	2.31	0.943	0.050	−0.980
NPS29	2	2.18	0.963	0.376	−0.824
NPS30	2	1.84	0.901	0.841	−0.162
NPS31	2	2.08	0.956	0.473	−0.771
NPS32	1	1.62	0.842	1.157	0.365
NPS33	2	2.13	0.912	0.324	−0.795
NPS34	2	1.80	0.923	0.855	−0.348
NPS35	2	2.26	0.975	0.360	−0.839
NPS36	2	1.82	0.932	0.955	−0.002
NPS37	2	1.85	0.961	0.840	−0.393
NPS38	2	1.94	0.922	0.719	−0.346
NPS39	2	1.88	0.946	0.809	−0.348
NPS40	2	1.98	0.949	0.614	−0.615
NPS41	2	1.85	0.899	0.784	−0.284
NPS42	2	1.82	0.859	0.731	−0.368
NPS43	2	2.08	0.995	0.417	−0.994

**Table 3 healthcare-09-01717-t003:** Rasch fit statistics for the Narcissistic Personality Scale.

	Item Description	Item–Total Correlation	Measure	S.E.	Infit MNSQ	Outfit MNSQ	DIF Contrast		
							Sex	Age	Educ
01	Your thoughts and opinions are better than those of others.	0.527	−0.43	0.08	1.00	0.99	0.05	−0.11	−0.09
02	You think that the world must remember you.	0.590	−0.22	0.07	1.03	1.06	0.09	0.21	−0.21
03	When someone causes you problems, you do not have to talk with them. However, you want those in charge to deal with the problem for you.	0.492	−0.28	0.07	1.14	1.37	−0.45	0.08	0.52
04	Most people do not congratulate or praise you as much as they should.	0.574	0.15	0.08	1.06	1.06	0.03	0.09	0.02
05	You deserve to receive care and special attention.	0.527	−0.15	0.08	1.10	1.20	−0.23	0.31	−0.35
06	Sometimes it becomes necessary to affect others to complete your job.	0.507	0.03	0.08	1.10	1.09	0.11	0.12	0.54
07	You are brave to criticize others when that certain person is insufficiently capable of performing the job.	0.562	−0.33	0.07	1.01	1.02	0.00	−0.32	0.24
08 *	Often some people are jealous of you.	0.561	0.2	0.08	1.04	1.08	−0.05	−0.26	0.66
09	You do not want to listen to those who are inferior to you.	0.599	0.19	0.07	0.98	0.98	−0.07	0.32	−0.15
10	You feel that you still have some charisma.	0.561	−1.00	0.07	1.03	1.03	−0.14	0.13	−0.58
11	People in general do not realize their own status.	0.477	−0.36	0.08	1.11	1.09	−0.08	0.22	0.17
12	When talking about your accomplishments, you deeply hope others to compliment you, even though you said you do not need so.	0.602	−0.37	0.07	0.92	0.90	−0.25	−0.11	0.00
13	You feel that you should deserve anything before others.	0.734	0.03	0.08	0.92	0.93	0.00	−0.07	−0.45
14	Some people who are not useful should not be retained.	0.597	0.32	0.08	1.00	0.97	0.26	0.3	−0.47
15	No matter where you are, there will always be someone who is one step ahead of you.	0.469	0.42	0.08	1.15	1.21	0.05	−0.21	0.07
16	Those around you are most generally inferior to you.	0.638	0.2	0.07	0.90	0.89	−0.06	0.00	0.00
17	Your actual rival is not just an ordinary/general person.	0.628	0.22	0.08	0.93	0.95	0.32	−0.57	0.49
18	Sometimes you have to lie to save your face.	0.596	0.17	0.08	0.91	0.94	0.06	0.00	−0.12
19	You think that you deserve good things, and no one can get in the way.	0.584	0.39	0.08	0.76	0.72	0.02	0.10	0.00
20	When someone within your authority does something wrong, you consider it unacceptable.	0.573	0.22	0.08	1.08	1.09	0.22	0.19	−0.17
21	Those around you are not as good as you. You do not think that you are good. Nonetheless, as a matter of fact, you are good.	0.558	−0.16	0.08	1.02	1.08	0.25	0.07	0.20
22	Some people have accused you of being selfish.	0.571	0.42	0.08	0.93	0.93	0.02	0.18	−0.18
23	You cannot accept when some people do not trust you.	0.519	−0.31	0.07	1.05	1.06	−0.27	−0.39	0.42
24	Many people approach you because you are a model of a successful person.	0.460	−0.23	0.08	1.12	1.24	0.08	−0.51	0.46
25 *	If your child is not as good as you expected, the child is not yours.	0.621	0.84	0.09	0.89	0.85	−0.16	0.28	−0.76
26	People presume to act equal to you.	0.616	0.04	0.08	0.91	0.90	0.23	−0.02	0.11

MNSQ = mean square, S.E. = standard error, DIF = differential item functioning, * DIF item.

## Data Availability

The datasets used and/or analyzed during the current study are available from the corresponding author upon reasonable request.

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
