# Peer review of "Developing and Validating the Narcissistic Personality Scale (NPS) among Older Thai Adults Using Rasch Analysis"

_healthcare, 2021, doi:10.3390/healthcare9121717_

Round 1

Reviewer 1 Report

The authors describe the construction of a questionnaire assessing narcistic personality disorder in Thai older adults using IRT methods. There is definitely a need for a psychometrically sound questionnaire with this measurement intention. The analyses are state of the art, including IRT measurement models and tests of differential item functioning. However, the description of the analyses could be improved (linguistically) to explain the rationale and procedure to the general reader. The issue that ICD-derived indicators tend to be too difficult for the general population is a frequent finding. Nevertheless, implications of this limitation should be discussed. Apart from that, I enjoyed very much reading this psychometrically sound contribution. A couple of, mostly minor, suggestions are mentioned below that could be addressed in a revision. 

- 88: How were the exclusion criteria assessed in this online study -- I assume by means of self-report? 

- 96: Why was the sample size determined for a factor analytic study? I think recommendations for polytomous IRT models tend to be a little higher, in particular when dimensionality is tested (e.g., Jiang, et al., 2016; doi: 10.3389/fpsyg.2016.00109). The actual number of participants would be informative in the Participants section.

- 98 f.: When NPD is conventionally assumed to have at least 9 separable facets, why did the authors not test a hierachical (bifactor or higher order) model by means of CFA? 

- 114: Some earlier contributions suggested that the content validity index should be >.80 (e.g., Lynn, 1986, doi: https://doi.org/10.1097/00006199-198611000-00017). However, I do not insist, as such cut-off values appear arbitrary. 

- 116: An n=34 appears small for Rasch analyses. The sample size of the "larger sample" in the subsequent study (l. 128) is not specified here. 

- 139-144: The list could be formatted differently to facilitate reading (e.g., with bullet points?) 

- 145 ff.: Paragraphs 2.4.2 and 2.4.2 should be rewritten for clarity (linguistically; the statistical criteria adopted by the authors make sense).

- 178: Possibly replace "patient ability" by "participant ability", following IRT conventions?

- Figure 1: The PRISMA flow chart does not offer much information beyond what is given in the text. 

- 251 ff.: Relevant issue such as that items tended to be too difficult to assess persons with low (i.e., normal) narcistic attitudes reliably should be discussed in the Limitations section.

- I suggest presenting Limitations first, followed by the Future Research section. This way, limitations can be discussed, then possible remedies presented afterwards. 

- Table 3 and the Appendix could be integrated into one table to save space. Then, the item wording would need to be given instead of short descriptions, and the 2 removed items have to be marked.

Author Response

Reviewer 1

1)- 88: How were the exclusion criteria assessed in this online study -- I assume by means of self-report? 

Response. Thank you for this query. We have asked and relied on the data the respondent provided. It is a limitation of such online survey in terms of reliability. However, we gain confidence about that by checking the internal consistency, item and person reliability provided.

2)- 96: Why was the sample size determined for a factor analytic study? I think recommendations for polytomous IRT models tend to be a little higher, in particular when dimensionality is tested (e.g., Jiang, et al., 2016; doi: 10.3389/fpsyg.2016.00109). The actual number of participants would be informative in the Participants section.

Response. Thank you for pointing this out, and your recommended paper. It was our real mistake. The sample size calculation used was relied on Linacre’s recommendation.  We have revised this part as follows.

As recommended by Linacre, a sample size between 30 and 200 could be sufficient for Rasch analysis (J.M. Linacre, 1994). A 30 sample could provide a statistically stable measure for the original draft of the questionnaire to reduce the items by identifying those misfitting. A sample of at least 200 was used to further test for the second set of the questionnaires based on Rasch’s required criteria.

(J.M. Linacre, 1994)

3)- 98 f.: When NPD is conventionally assumed to have at least 9 separable facets, why did the authors not test a hierachical (bifactor or higher order) model by means of CFA? 

Response. Thank you for these thoughtful questions. We totally agree that bifactor model is another way to demonstrate how sufficiently unidimensional the scale is. However, NPS is in the process of developing to secure the number of items to be remained in the scale. This process would be better to use Rasch analysis as it can provide us, not only the validity of the construct, but also a useful plan., e.g., how well the NPS is targeted with this nonclinical older people, what kind of item should be added.

4)- 114: Some earlier contributions suggested that the content validity index should be >.80 (e.g., Lynn, 1986, doi: https://doi.org/10.1097/00006199-198611000-00017). However, I do not insist, as such cut-off values appear arbitrary. 

Response: Thank you for this information. We have revised it accordingly.

5)- 116: An n=34 appears small for Rasch analyses. The sample size of the "larger sample" in the subsequent study (l. 128) is not specified here. 

Response. We have revised this as shown in Q2.

6)- 139-144: The list could be formatted differently to facilitate reading (e.g., with bullet points?) 

Response: Thank you for your suggestion but we have removed this part by the suggestion of another reviewer to make it section more concise.

7)- 145 ff.: Paragraphs 2.4.2 and 2.4.2 should be rewritten for clarity (linguistically; the statistical criteria adopted by the authors make sense).

Response. We have an English native speaker edited these parts again.

2.4.1. Examining the fit between the data and the model

Chi-square fit statistics were calculated to indicate how well the empirical data fit the Rasch model. These fit statistics are the outlier-sensitive fit statistics mean square (outfit MNSQ) and information-weighted fit statistics mean square (infit MNSQ). The outfit statistic is more sensitive to outliers, whereas the infit MNSQ is sensitive to unexpected responses near the person’s ability level. The expected infit or outfit mean square values are 1.0; MNSQ >2.0 indicating distorting or degrading the measurement system; MNSQ, 1.5 - 2.0 indicating unproductive for constructing measurement, but not degrading. An item with infit or outfit MnSq of the 0.7–1.5 range was considered a misfit [18].

Misfit items and person performance patterns that are too erratic to meet the Rasch model probabilistic expectations are misfitting. Misfitting items and persons’ performances degrade measures.

2.4.2. Dimensionality examination

Principal component analysis of the residual was used to identify the Rasch dimension, the only dimension in the data. However, a secondary dimension suggested the unexplained variance of the 1st contrast >2 eigen values (at least 3). However, if the disattenuated correlation between person measures was more than 0.70, it may merely be due to noise or an idiosyncratic item [19]. Local independence is a basic assumption of item response regarding theory models, in which the observed items are presumably independent of each other given an individual score on the latent variable(s). This is evaluated by determining a positive correlation of the size of the standardized residuals for two items (or persons). An acceptable correlation is less than 0.2, denoting that the pair of items is not duplicate or shared in the construct [20].

8)- 178: Possibly replace "patient ability" by "participant ability", following IRT conventions?

Response: Thank you for informing an updated guideline. We have replaced ‘patient’ by ‘participant wherever appropriate. However, we could not the word ‘patient ability’ in the revised one.

9)- Figure 1: The PRISMA flow chart does not offer much information beyond what is given in the text. 

Response: Thank you for this comment. We have removed the flow chart.

10)- 251 ff.: Relevant issue such as that items tended to be too difficult to assess persons with low (i.e., normal) narcistic attitudes reliably should be discussed in the Limitations section.

Response:  Thank you for this point. We have added this limitation as follows.

Assessing people with low (i.e., normal) narcistic attitudes reliably is difficult because assessment is based on the DSM criteria, so easier items that represent these criteria should be added and tested in further study.

11)- I suggest presenting Limitations first, followed by the Future Research section. This way, limitations can be discussed, then possible remedies presented afterwards. 

Response: Thank you very much. We totally agree with this suggestion.

12)- Table 3 and the Appendix could be integrated into one table to save space. Then, the item wording would need to be given instead of short descriptions, and the 2 removed items have to be marked.

Response: Thank you for this suggestion. We absolutely agree with that and revise accordingly. The appendix is removed.

Reviewer 2 Report

I am grateful to the editors for the opportunity to collaborate as a reviewer in the Healthcare. I would also like to congratulate the authors of the manuscript "Developing and Validating the Narcissistic Personality Scale (NPS) among Thai Older Adults Using Rasch Analysis", for the effort made in their study.

I recommend improving the following aspects:  

  1. The inclusion criteria should be included.
  2. Modify the Statistical Analysis section by eliminating unnecessary information. This section should be adequately summarised.
  3. Figure 1 does not provide information; its use in systematic review articles is more relevant. Its elimination is recommended.
  4. In table two, the minimum and maximum columns do not provide information, as it is the same result in all the items. They should be deleted.
  5. The item-total correlation of all items should be included.
  6. Revise line 215 of the manuscript, it looks like a title.
  7. It is recommended to include the omega coefficient together with the Cronbach's alpha.
  8. As the validation of a questionnaire, it is recommended to carry out the exploratory factor analysis (including the necessary assumptions for its performance such as KMO, Bartlett's test of sphericity and determinant), as well as the factorial invariance between men and women.
  9. The discussion should be modified to include the information requested in this review.

Author Response

Reviewer 2

  1. The inclusion criteria should be included.

Response: Thank you for your suggestion. In our proposal we have written the inclusion criteria as described below.

Inclusion Criteria

  1. Age ≥60 years
  2. Having ability to talk, write, and communicate well in Thai.
  3. Completing the questionnaire via a device or a computer or being able to read the questionnaire by themselves.

Therefore, we have written “The participants were Thai older adults residing in Thailand, aged 60 years and higher, able to read and write in Thai, and able to access the Internet.” in our manuscript. Hence, we haven’t revised this information.

  1. Modify the Statistical Analysis section by eliminating unnecessary information. This section should be adequately summarized.

Response: Thank you for you suggestion. We have revised the Statistical analysis section as shown below.

  • Statistical analysis

For sociodemographic and descriptive statistics, mean, SD, and frequency were used. Internal consistency of the instrument using IBM SPSS, Version 22, was determined using Cronbach’s alpha, for which a value >0.8 was considered acceptable.

Rasch analysis

The Polytomous Rasch rating scale model using the Winsteps Measurement Software (Winsteps, Rasch Measurement, Version 5.1.5.0, Chicago, IL, USA) verified the analyses.

2.4.1 Examining the fit between the data and the model

Chi-square fit statistics were calculated to indicate how well the empirical data fit the Rasch model. These fit statistics are the outlier-sensitive fit statistics mean square (outfit MNSQ) and information-weighted fit statistics mean square (infit MNSQ). The outfit statistic is more sensitive to outliers, whereas the infit MNSQ is sensitive to unexpected responses near the person’s ability level. The expected infit or outfit mean square values are 1.0; MNSQ >2.0 indicating distorting or degrading the measurement system; MNSQ, 1.5 - 2.0 indicating unproductive for constructing measurement, but not degrading. An item with infit or outfit MnSq beyond the 0.7–1.5 range was considered a misfit (Wright BD, 1994).

2.4.2 Dimensionality examination

Principal component analysis of the residual was used to identify the Rasch dimension, the only dimension in the data. However, a secondary dimension suggested the unexplained variance of the 1st contrast >2 eigen values (at least 3). However, if the disattenuated correlation between person measures was more than 0.70, it may merely have been due to noise or an idiosyncratic item (J. M. Linacre, 2017). Local independence is a basic assumption of item response theory models, in which the observed items are presumably independent of each other given an individual score on the latent variable(s). This is evaluated by determining a positive value of the correlation of size of the standardized residuals for two items (or persons). An acceptable correlation is less than 0.2, denoting that the pair of items is not duplicated or shared in the construct (Christensen, Oernboel, Zatzick, & Russo, 2017).

2.4.3 Reliability and separation indices

Reliability is estimated both for persons and for items. Person reliability in Rasch analysis is comparable to Cronbach’s alpha. The higher the separation index, the better the instrument is able to differentiate person ability and item difficulty. Low person reliability indicates a narrow range of person measures, or a small number of items. Low item reliability denotes that the sample is insufficient to locate the items on the latent variable correctly.

2.4.4 Wright Map

Well-targeting between person and item is denoted by the mean location for the person and should be around zero logits. It has been suggested that the difference between the mean value of the mean person measure should be within one logit. Floor or ceiling effects could also be examined visually using the map.

2.4.5 Differential item functioning

We tested the differential item functioning (DIF) across sex, age, and education. Both statistical test and DIF contrast were used, and a DIF contrast >0.64 indicated a substantial DIF (J. M. Linacre, 2017).

2.4.6 Category function

Category functioning is assessed by determining category frequencies, mean measures, thresholds, and category fit statistics (Tennant & Conaghan, 2007). The items of the NPS have four categories. At least ten responses per category is recommended for stable rating scale–structure threshold parameter estimates (Tennant & Conaghan, 2007). The mean measures and the thresholds should increase from lower to higher categories. An increase between 1.4 and 5 logits denotes suitable threshold.

  1. Figure 1 does not provide information; its use in systematic review articles is more relevant. Its elimination is recommended.

Response: Thank you for your suggestion. We removed Figure 1 and related content in the text. We have also revised Figures 2 and 3 and their relevant contents in the text to be Figures 1 and 2, respectively.

  1. In table two, the minimum and maximum columns do not provide information, as it is the same result in all the items. They should be deleted.

Response: Thank you for your suggestion. We have deleted them.

  1. The item-total correlation of all items should be included.

Response: Thank you for your suggestion. We have added the correlation of all items in the Table 3. We have added text, “The Item-Total correlation of 26 items ranged from 0.460-0.734.” in 3.3 please see page 8.

  1. Revise line 215 of the manuscript, it looks like a title.

Response: Thank you for your suggestion. The subheading, “Fit statistics” - line 215, was omitted.

  1. It is recommended to include the omega coefficient together with the Cronbach's alpha.

Response: We have added omega coefficient together with the Cronbach's alpha as suggested. It now reads, “The McDonald’s w coefficient was .933.”

  1. As the validation of a questionnaire, it is recommended to carry out the exploratory factor analysis (including the necessary assumptions for its performance such as KMO, Bartlett's test of sphericity and determinant), as well as the factorial invariance between men and women.

Response: Thank you for this suggestion but we originally intended to use Rasch analysis based on Rasch measurement theory instead of factor analysis (Classic test theory-CTT) because in this study we are developing a scale according to Rasch measurement model. Despite the fact that two methods of analysis may yield the same results, CTT (factor analysis) aims to find the best model to fit the data. On the other hand, we would like to fit our data to the (Rasch) measurement model. Rasch analysis has covered all required analyses to provide a construct validity test for a newly developed scale. In addition, the functions of category probability and Wright Map allows us, as researchers, to see whether the scale is appropriate for the targeted population or not, as well as to help us decide on improving the scale in the future. In addition, based on Rasch measurement theory, local dependence is not encouraged, whereas it might not constitute an issue in CTT (1-3). Overall, we believe that Rasch analysis is a sufficient analysis based on measurement theory, to provide us the required results, e.g., dimensionality, measurement invariance or differential item functioning. Although, some investigator would like to use both methods to compare the results (4), that would be the case if we used a different method. However, at the present, we would like to focus on presenting how the NPS was developed. Therefore it would be more appropriate not to overwhelm or probably confuse the readers with considerable analysis of results for such a newly developed scale. However, we agree with the reviewer that the NPS could be compared for the results using a different method of analysis in the future with a new sample.

  • Tennant A, Conaghan PG. The Rasch measurement model in rheumatology: what is it and why use it? When should it be applied, and what should one look for in a Rasch paper?Arthritis Care Res. 2007;2007:57. doi:10.1002/art.23108
  • Bond TG. Applying the Rasch Model: Fundamental Measurement in the Human Sciences/Authored by Trevor G. Bond and Christine M. Fox. Routledge, Taylor & Francis Group; 2015. 
  • Construct Validity and Differential Item Functioning of the PHQ-9 Among Health Care Workers: Rasch Analysis Approach - PubMed (nih.gov)
  • Evaluating hierarchical items of the geriatric depression scale through factor analysis and item response theory - PubMed (nih.gov)

  1. The discussion should be modified to include the information requested in this review.

Response: Please see Q8

Reviewer 3 Report

I would like to thank the aAuthors for the opportunity to read their article. Below, I present my comments and questions.

Introduction

Line 27 "This personality trait worsens when"

It would be more accurate to say that a personality trait is increasing rather than worsens. It is worth making no evaluation but description in scientific texts.   

There is no presentation of the causes of narcissism in the introduction. 

The types of narcissism should also be mentioned (e.g. Vulnerability and Grandiosity). 

Research on narcissism in Thailand should be referenced in the article: 

https://doi.org/10.1016/j.paid.2019.05.009

https://doi.org/10.1016/j.paid.2017.08.024

Line 60

They constitute widely used measurements of narcissism [12,13] despite the fact that they are unacceptable in terms of model fit and factorial valid-61 ity. More importantly, they are not specific to old age populations

What does it mean that they are "are unacceptable"? please elaborate on it. 

3.3. Reliability

The McDonald's ω index should be added. 

Why were other narcissism questionnaires not used to measure criterion validity? 

In the discussion, reference can be made to other studies conducted on the elderly and in Thailand. 

10.1080/13607863.2020.1725802  or  https://doi.org/10.1093/geroni/igaa057

6. Strength and limitation of the study

In the limitations of the research, it should be noted that not all people over 60 in Thailand have access to the Internet. They may differ from the study group by education, social status, income, place of residence, and the use of social media. Probably different results could be obtained if they were tested with the traditional paper-and-pencil method. 

Author Response

Reviewer 3

  1. Line 27 "This personality trait worsens when"

It would be more accurate to say that a personality trait is increasing rather than worsens. It is worth making no evaluation but description in scientific texts.   

Response:

Thank you. We have replaced “worsens” with “is increasing” as suggested. Please see page 1.

  1. There is no presentation of the causes of narcissism in the introduction.

Response: Thank you for your suggestion. We have added text, “The etiology of narcissistic personality disorder is multifactorial. Genetic predisposing is suggested to be one of the causes in some studies (Luo, Cai, & Song, 2014). Aggression, reduced tolerance to distress, and dysfunctional affect regulation is prominent among individuals with NPD (Minnix, Romero, Joiner, & Weinberg, 2007). Developmental experiences, negative in nature, being rejected as a child, and a fragile ego during early childhood may have been linked \ to the occurrence of NPD in adulthood (Kernberg, 1989; Otway & Vignoles, 2006), to the introduction section.

  1. The types of narcissism should also be mentioned (e.g. Vulnerability and Grandiosity).

Response: Thank you for your suggestion. We have added text “Evidence has suggested that the two distinct dimensions of narcissism, are often referred to as grandiose and vulnerable narcissism (Kraus, Boswell, Wright, Castonguay, & Pincus, 2010; Miller et al., 2011; Miller et al., 2018).

to the introduction section.

  1. Research on narcissism in Thailand should be referenced in the article:

https://doi.org/10.1016/j.paid.2019.05.009

https://doi.org/10.1016/j.paid.2017.08.024

 Response: We have added some sentences according to the references as follows,

Studies on narcissistic personality among Thais are scarce. One conducted among young Thai athletes revealed that narcissistic admiration was positively correlated with self and coach-ratings of mental toughness, whereas narcissistic rivalry was negatively correlated with self and coach-ratings of mental toughness (Manley, Jarukasemthawee, & Pisitsungkagarn, 2019).

  1. Line 60

They constitute widely used measurements of narcissism [12,13] despite the fact that they are unacceptable in terms of model fit and factorial validity. More importantly, they are not specific to old age populations

What does it mean that they are "are unacceptable"? please elaborate on it. 

Response: Thank you for your suggestion. We have revised the text to, “however, they are not specific to old age populations”.

  1. 3. Reliability

The McDonald's ω index should be added. 

Response: Thank you. We have added as suggested.

  1. Why were other narcissism questionnaires not used to measure criterion validity?

Response: Thank you for your comment. In this paper, we focused on presenting the construct validity using Rasch analysis, and plan to do other validity tests in our next study.

  1. In the discussion, reference can be made to other studies conducted on the elderly and in Thailand.

10.1080/13607863.2020.1725802  or  https://doi.org/10.1093/geroni/igaa057

 Response: We have added some sentences according to the references as follows,

                                                Introduction

Other personality disorders such as antisocial personality disorder, borderline personality disorder, histrionic personality disorder, and schizotypal personality disorder are also commonly comorbid among people with NPD (Ronningstam, 2011). A study showed that both Vulnerability, and Grandiosity types of narcissism were significantly correlated with various personality disorders, except for Schizotypal PD, as well as the personality traits of Negative Affect and Antagonism (Stone, Segal, & Krus, 2021).

Discussion

       Culture plays important roles in the character of narcissism that would influence which item is to be selected for the questionnaire. A study comparing narcissism between western (Germany) and eastern (Japan) cultures documented that grandiose narcissism was prevalent in western whereas vulnerable narcissism was prevalent in the eastern countries. The entitlement factor was assumed to be equivalent to the construct whereas the relationship between narcissism and mental health problems differ (Jauk, Breyer, Kanske, & Wakabayashi, 2021). We assumed that this cultural incongruency-hypothesis concerning narcissism and mental health might be related to the different components of specific features of narcissism between the two cultures. A study conducted in Thai culture revealed how people express themselves through social media such as Selfies on Facebook, implying a narcissistic tendency. Older people with narcissistic personality might show their photos with family members expressing gratitude or caring to inflate their self-esteem (Isaranon, 2019).

  1. Strength and limitation of the study

In the limitations of the research, it should be noted that not all people over 60 in Thailand have access to the Internet. They may differ from the study group by education, social status, income, place of residence, and the use of social media. Probably different results could be obtained if they were tested with the traditional paper-and-pencil method. 

Response: Thank you for your suggestion. We have revised this part by adding text, “In addition, some older people in Thailand may not be able to access internet, so different results could probably be obtained if they were tested with the traditional paper-and-pencil method due to the difference of education, social status, income, place of residence, and the use of social media between those who were able to access and those who were not.”.
